# Street Gang Intervention: Review and Good Lives Extension

**Jaimee Mallion** [1,*] **and Jane Wood** [2]

[1] School of Psychology, London South Bank University, London SE1 6LN, UK
[2] School of Psychology, University of Kent, Kent CT2 7NZ, UK; J.L.Wood@kent.ac.uk
* Correspondence: mallionj@lsbu.ac.uk

**Abstract:** Tackling street gangs has recently been highlighted as a priority for public health. In this paper, the four components of a public health approach were reviewed: (1) surveillance, (2) identifying risk and protective factors, (3) developing and evaluating interventions at primary prevention, secondary prevention, and tertiary intervention stages, and (4) implementation of evidence-based programs. Findings regarding the effectiveness of prevention and intervention programs for street gang members were mixed, with unclear goals/objectives, limited theoretical foundation, and a lack of consistency in program implementation impeding effectiveness at reducing street gang involvement. This paper proposes that the Good Lives Model (GLM), a strengths-based framework for offender rehabilitation, provides an innovative approach to street gang intervention. Utilizing approach-goals, the GLM assumes that improving an individual's internal skills and external opportunities will reduce the need to become involved in street gangs. Wrapping the GLM framework around current evidence-based interventions (e.g., Functional Family Therapy) increases client engagement and motivation to change, which is notably poor amongst those at risk of, or involved in, street gangs.

**Keywords:** street gangs; public health; Good Lives Model; intervention; prevention

## 1. Introduction

Street gangs are a growing problem internationally, with countries including the UK, USA, Sweden, China, and the Netherlands reporting a marked increase in street gang membership (e.g., Chui and Khiatani 2018; Roks and Densley 2020; Rostami 2017). In the UK alone, the number of street gang affiliated youths has seen a dramatic increase over a five-year period. The Children's Commissioner (2017) approximated that in 2013/14, 46,000 young people were either directly gang-involved or knew a street gang member. By 2019 this figure had increased to 27,000 full street gang members, 60,000 affiliates, and a further 313,000 youths who knew a street gang member (Children's Commissioner 2019). Similar increases have been seen in the USA, with a 40.83% growth in the number of different street gangs between 2002 and 2012 (National Gang Center 2020). As such, the World Health Organization (World Health Organization 2020) has highlighted youth violence, including street gang membership, as a global public health problem that requires an immediate international response.

Street gang membership is associated with increased perpetration of illegal activities, particularly serious and violent offences (Pyrooz et al. 2016), with this relationship stable across time, place, and definitions of street gangs (Dong and Krohn 2016). As such, street gangs are responsible for causing heightened levels of fear and victimization amongst members of their community (Howell 2007). In addition, street gang involvement has adverse health, welfare, and economic consequences for individual members, which persist long after disengagement (Connolly and Jackson 2019; Petering 2016). For instance, longitudinal research identified that adults who belonged to a street gang during adolescence experienced more mental and physical health issues than their non-gang

counterparts (Gilman et al. 2014). Adolescent street gang members also experience more economic hardship during adulthood than their non-gang peers, with higher rates of unemployment and reliance on welfare benefits or illicit income (Krohn et al. 2011). Furthermore, street gang involvement during adolescence has a detrimental effect on the development of long-term stable family relationships, with former members more likely to engage in intimate partner violence and child maltreatment (Augustyn et al. 2014).

Considering these long-term and wide-ranging effects of street gang membership, it is unsurprising that there has been a proliferation of prevention and intervention programs developed and implemented world-wide. Although literature is beginning to emerge which suggests some of these are effective programs at reducing street gang involvement, there remains a paucity of reliable evidence to date. Highlighted by Wong et al. (2011). such programs often suffer from a lack of theoretical foundation (McGloin and Decker 2010), clear goals and objectives (Klein and Maxson 2006), and methodologically sound evaluation (Curry 2010). These factors are associated with an increased risk of harmful outcomes for program participants (Welsh and Rocque 2014), including negative labeling and heightened rates of recidivism (Petrosino et al. 2010). Thus, discovering "what works" in street gang prevention and intervention is essential.

A public health approach to street gang membership has recently been suggested (Gebo 2016), which could guide the development of effective prevention and intervention strategies. WHO (Krug et al. 2002) suggests four key elements for a public health approach, including: (1) surveillance, (2) identifying risk and protective factors, (3) developing and evaluating interventions, and (4) implementation. See Figure 1 for an overview of each of these elements in relation to street gang prevention and intervention. Using a public health approach, street gang intervention occurs across three levels (Conaglen and Gallimore 2014): primary prevention (early intervention approaches prior to initiation of street gang involvement), secondary prevention (interventions specifically for individuals at-risk of street gang involvement), and tertiary prevention (long-term rehabilitation strategies for those who have engaged in street gangs). In addition, public health interventions can be universally implemented (aimed at the general population), selected (targeted towards those at-risk of street gang involvement), or indicated (targeted specifically at street gang members).

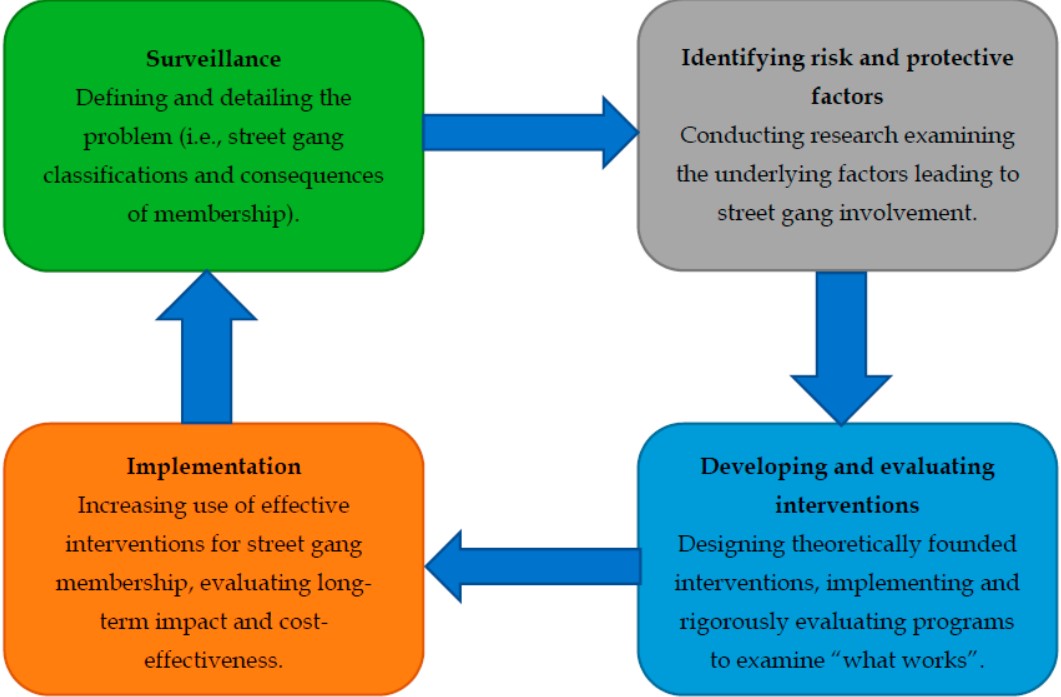

**Figure 1.** WHO's public health approach to violence prevention (Krug et al. 2002), adapted for street gang intervention.

Public health approaches have seen a number of successes in reducing behaviors related to street gang membership (e.g., substance misuse, child maltreatment and youth violence; HM Government 2019; Pickering and Sanders 2015; Public Health England 2015). However, research is limited regarding the effectiveness of interventions for street gang members (McDaniel et al. 2014). The aim of this paper is to narratively summarize and evaluate existing street gang prevention and intervention programs, within a public health approach. Aspects of the public health approach will be outlined in relation to street gang membership, including: (1) surveillance (i.e., street gang definitions), (2) risk and protective factors, (3) current street gang prevention and intervention programs (including primary, secondary, and tertiary interventions). Furthermore, this paper will examine how a novel approach to offender rehabilitation, termed the Good Lives Model (Ward and Fortune 2013), could be used as a framework to guide street gang intervention.

## 2. Surveillance

Surveillance is a core aspect of a public health approach, which informs the development and implementation of prevention and intervention programs (Richards et al. 2017). Surveillance involves establishing clear definitions regarding the population of interest (i.e., street gang members), enabling the identification of both those in need of intervention and the associated risk factors (Department of Health 2012). By implementing surveillance measures, such as analyzing knife crime and criminal convictions data, the extent of the problem in society on a local, national, and international scale can be recognized (World Health Organization 2010). Ongoing monitoring enables any changes in the patterns or frequencies of behavior to be quickly identified and disseminated to intervention providers, informing the decision-making process (Public Health England 2017).

*Street Gang Definition*

The definition of a street gang member has been a matter of ongoing debate amongst academics, policy-makers, and stakeholders for decades (e.g., Esbensen and Maxson 2012). To date, no single, standardized definition of a street gang has been agreed. The ambiguity surrounding the definition of a street gang has serious consequences for the development of effective prevention and intervention strategies. As Melde (2016, p. 160) explains, "you cannot manage what you cannot measure". Without a reliable and valid definition, stakeholders are unable to accurately measure the rates of street gang members and street gang-related offending. In addition, a lack of clear definitional criteria prevents an assessment of the short- and long-term impact of prevention and intervention strategies on street gang dynamics (Melde 2016).

To overcome this, stakeholders often devise their own street gang definition, which allows them to undertake surveillance procedures and see the impact that prevention and intervention strategies have on the local area. However, definitions of a street gang often vary widely from one region to the next (Gilbertson and Malinksi 2005). For instance, each jurisdiction in the USA has its own definition of a street gang and what constitutes a street gang-related offence (for a summary of definitions, see National Gang Center 2016). Despite attempting to measure the same phenomenon, by using different definitions a large disparity is likely to emerge in the estimates of street gang members and rates of street gang-related offending between areas. Dependent on the definition used, an over-identification (incorrectly identifying an issue as related to street gang membership, when it is not) or under-identification (incorrectly identifying an issue as unrelated to street gangs, when it is) of street gang members and street gang-related offending can occur (Joseph and Gunter 2011). As such, prevention and intervention strategies for street gang members may be offered to too few or too many in the local area. Furthermore, the differences in definitions used mean the generalizability of any prevention and intervention strategies across areas is limited.

One method of identifying street gang members is through self-nomination, whereby stakeholders simply ask individuals "are you currently in a gang?" (Esbensen et al. 2011). Past research has found self-nomination to be a valid and effective method of identifying street gang members (e.g., Decker et al.

2014; Esbensen et al. 2001; Matsuda et al. 2012). In addition, self-nomination of street gang membership is associated with heightened levels of violent crime (Melde et al. 2016), which is consistent with the extensive research suggesting street gang members are more likely to commit serious and violent offences than their non-gang counterparts (Melde and Esbensen 2013). However, self-nomination relies on the individuals' willingness to respond honestly, which could be reduced due to the negative impact of disclosing street gang membership (e.g., risk of incarceration or retaliation from street gang peers). Critically, self-nomination is dependent upon an individual's subjective understanding and interpretation of the term 'gang' (Tonks and Stephenson 2018). As public health surveillance requires street gang members to be identified by an objective party, self-nomination methods would not be appropriate.

The Eurogang Network, a group of the world's leading street gang researchers, attempted to establish a standardized definition of a street gang, which would allow cross-national comparative research and surveillance (Klein and Maxson 2006). According to the Eurogang definition, a street gang is a "durable, street-oriented youth group whose involvement in illegal activity is part of their group identity" (Weerman et al. 2009, p. 20). Specifically, the group must: (1) include more than three people, (2) last longer than three months, (3) be street-orientated, (4) be acceptive of illegal activities, and (5) engage in illegal activities together (Matsuda et al. 2012). Critically, the Eurogang definition does not require an individual to self-nominate in order to be classed as a street gang member. The Eurogang Network avoids using the term 'gang' due to its emotive nature, instead preferring "troublesome youth group" (Esbensen and Weerman 2005).

Although the Eurogang definition is increasingly adhered to in academic research, policy-makers and stakeholders are resistant to its use. For instance, stakeholders have suggested that avoidance of the term 'gang' reduces their ability to effectively distinguish between a street gang and a group of individuals who happen to commit offences together (Centre for Social Justice 2009; Pearce and Pitts 2011). Supporting this, researchers have found that the Eurogang definition leads to an over-categorization of groups as street gangs (e.g., illegal ravers, peer groups who consume drugs; Medina et al. 2013). Aldridge et al. (2012) suggests this is due to a lack of defining criteria concerning street gang members engagement in violent crime. Despite typically being used in academia as a self-report measure, the Eurogang criteria are observable (i.e., stakeholders can see whether a young person is in a large street-based group, committing crimes), enabling surveillance measures for identifying and monitoring street gangs (Melde 2016). To support consistency across surveillance measures and intervention provision, it is recommended that the Eurogang definition is used to guide a public health approach to street gangs.

## 3. Risk and Protective Factors

A public health approach involves developing an understanding of the causes of street gang membership (Local Government Association 2018). This takes two forms, with the identification of risk factors (increasing the likelihood of street gang involvement) and protective factors (reducing the likelihood of street gang involvement). By establishing a framework of risk and protective factors, this informs the development of prevention and intervention strategies aimed at reducing involvement in street gangs. To date, focus has been placed on identifying the risk factors for street gang membership, with a paucity of research on the protective factors (McDaniel 2012). This section will outline the risk and protective factors for street gang membership that have been identified.

### 3.1. Risk Factors for Street Gang Membership

Past research has demonstrated that there are a wide range of risk factors robustly associated with street gang membership. These span each of the five major risk factor domains: the individual, peers, family, school, and community (O'Brien et al. 2013). The risk factors which have been related to street gang membership across each of these domains are summarized in Table 1. Critically, Klein and Maxson (2006) noted that a number of risk factors for street gang membership are supported by weak or

inconclusive evidence. However, it must be considered that the evidence-base for street gang-related risk factors has rapidly grown since Klein and Maxson's (2006) suggestions. Yet, to complicate matters further, research has also suggested differences in risk factors within street gangs. Specifically, core street gang members (i.e., those that self-identify as street gang members) are more likely than peripheral members (i.e., those that engage in street gang crime, but do not self-identify as members) to have early exposure to deviant peer groups, low impulse control, poor academic attainment, and endorse antisocial attitudes (e.g., Alleyne and Wood 2010; Esbensen et al. 2001; Klein 1995; Melde et al. 2011). This suggests peripheral and core street gang members have different needs that require targeting in intervention programs.

The presence of a risk factor does not determine that an individual will join a street gang. Indeed, many of the risk factors for street gang membership also predict other deviant behaviors (e.g., general delinquency and violence; Decker et al. 2013). However, the more risk factors the individual experiences, the higher the likelihood that they will engage in a street gang, beyond any other deviant behavior (Melde et al. 2011). Supporting this accumulative effect, Esbensen et al. (2010) found 11 or more risk factors were experienced by 52% of street gang members, compared with 36% of violent offenders. Street gang members are also more likely to concurrently experience risk factors in each of the major domains than their non-gang counterparts (Thornberry et al. 2003). This suggests that prevention and intervention strategies need to address numerous risk factors across all domains (Howell 2010).

### 3.2. Protective Factors for Street Gang Membership

In areas with a high presence of street gangs, over 75% of young people successfully avoid becoming members (Howell 2012). This is despite experiencing similar risk factors to those who engage in street gangs, particularly across the school and community domains. As suggested above, individuals who circumvent street gangs may not have accumulated as a high a number of risk factors as those that do become members. Alternatively, these individuals may experience more protective factors than those that do become affiliated with a street gang. In challenging environments, where it may not be possible to remove or reduce all risk factors, focusing on adding protective factors could decrease engagement in street gangs (Howell and Egley 2005).

However, with research predominantly focusing on the risk factors of street gang members, the protective factors have been neglected. The protective factors that have been identified so far span the individual, family, peer, and school domains (for a full summary, see Table 1). Regarding the individual, protective factors for an at-risk young person include having effective coping strategies, high emotional competence, and good social skills (Katz and Fox 2010; Lenzi et al. 2018; McDaniel 2012). For the family domain, protective factors include strong parental monitoring, cohesiveness within the family, and positive parental attachment (Li et al. 2002; Maxson et al. 1998). Interaction with prosocial peer groups is a protective factor within the peer domain (Katz and Fox 2010). Positive child-teacher relationships, clear familial expectations regarding schooling, and an individual's commitment to education are all protective factors in the school domain (Stoiber and Good 1998; Thornberry 2001). Little is known regarding the protective factors for street gang membership in the community domain. Future research examining protective factors is essential, particularly as strength-based approaches to offender rehabilitation have suggested that focusing on these could improve prosocial behavior in street gang members (O'Brien et al. 2013; Whitehead et al. 2007).

**Table 1.** Examples of risk factors for street gang membership, according to domain.

| Domain | Risk Factors | Protective Factors |
|---|---|---|
| Individual | Offence supportive cognitions *, negative life experiences *, low self-esteem, internalizing behaviors, externalizing behaviors *, impulsivity, lack of participation in prosocial activities, mental health issues (e.g., Post-Traumatic Stress Disorder, anxiety), negative attitudes towards the future, substance misuse, low empathy, high callous-unemotional traits, low trait emotional intelligence, moral disengagement, negative attitudes towards the police, hyperactivity, poor interpersonal skills, and anger rumination. | Effective coping strategies, high emotional competence, emotion regulation skills, resilient termperament, future orientation, impulse control, low ADHD symptomology, high self-esteem, intolerant attitude towards antisocial behavior, and belief in moral order |
| Peers | Negative peer influence *, association with delinquent peer group, victim or perpetrator of bullying, alienation from prosocial peers, strong emotional connection to delinquent peers, prioritizing social identity, and peers' substance misuse. | Interaction with prosocial peer groups, strong social skills, low peer delinquency, and prosocial bonding |
| Family | Poor parental supervision * and monitoring *, lack of attachment to parents, family involvement in street gangs, family involvement in crime, delinquent siblings, hostile family environment, parental substance misuse, inconsistent discipline, low familial socioeconomic status, single-parent households, childhood maltreatment, and running away from home. | Strong parental monitoring, control and supervision, parental warmth, cohesiveness within the family, positive parental attachment, stable family structure, and low levels of parent-child conflict |
| School | Poor academic attainment, lack of commitment to education, lack of aspirations, unsafe school environment, suspension/exclusion, truancy, inconsistent discipline, victimization at school, inadequate teaching, negative relationships with staff, and difficult transitions between schools. | Positive child-teacher relationships, clear familial expectations regarding schooling, personal commitment to education, positive role models, fair treatment from teachers, safe evironment, connectedness, regular school participation, and academic achievement |
| Community | Disorganized neighborhood, high rates of crime, exposure to street gangs and violence, availability of firearms, poverty, lack of community resources, and experiencing unsafe environments. | Opportunities for prosocial involvement, positive community role models, perceived neighborhood safety, and low economic deprivation |

Sources include: Home Office (2015), Lenzi et al. (2018), Mallion and Wood (2018), Melde et al. (2011), Merrin et al. (2015), O'Brien et al. (2013), Raby and Jones (2016), and Smith et al. (2019). * Risk factors identified by Klein and Maxson (2006) as having a robust evidence-base.

## 4. Current Approaches to Street Gang Intervention

Street gang membership has typically been targeted through the criminal justice system, including the imposition of street gang injunctions (behaviors or activities of the street gang member are prohibited, such as going to certain areas; HM Government 2016). Whilst research has demonstrated reductions in reoffending rates by recipients of street gang injunctions (Carr et al. 2017), long-term negative effects have also been identified (e.g., reduced opportunities for education and employment, and less access to prosocial networks; Swan and Bates 2017). However, there has been a recent growth in prevention and intervention programs which are psychologically-informed (e.g., O'Connor and Waddell 2015). These programs have more positive long-term outcomes, for both the individual and the community, than criminal justice approaches (Howell 2010), and fit well within a public health framework. This section will outline current approaches to street gang prevention and intervention, across three levels (primary, secondary, and tertiary).

### 4.1. Primary Prevention

In a public health approach, it is assumed that given the right conditions, any young person could be drawn towards joining a street gang (Gravel et al. 2013). As such, by using a universal approach, primary prevention strategies attempt to protect all young people from engaging in adverse behaviors (such as violence and street gang membership), by reducing risk and increasing protective factors (Gebo 2016). Primary prevention strategies include the provision of services which aim to reach and support a whole community. They are typically delivered via local schools, community outreach, and faith-based organizations (Wyrick 2006). These include ensuring equal access to education, employment, and housing, and improving the community space (i.e., cleaning communal areas and better lighting). Wyrick (2006) suggests these primary prevention strategies enhance community mobilization, which reduces engagement in street gangs.

Primary prevention strategies are commonly implemented in schools, as it is easy to reach a large number of young people prior to the onset of any deviant or delinquent behavior. One of the leading schools-based primary prevention programs for street gang membership is the Gang Resistance and Education Training Program (G.R.E.A.T; Esbensen et al. 2001; Esbensen et al. 2002). G.R.E.A.T is delivered by law enforcement officers to middle school pupils, aged 11–13 years, in the United States. The original version of G.R.E.A.T targeted risk factors not specific to street gang membership, including low self-esteem and unsafe schools (Klein and Maxson 2006). Despite program completers having more pro-social peers, negative attitudes towards street gangs, and fewer risk-taking behaviors, no difference was found between program recipients and non-recipients on levels of delinquency, violence, or street gang involvement (Esbensen et al. 2001).

As such, G.R.E.A.T underwent substantial changes, with the new curriculum comprising of 13 sessions targeting risk and protective factors specific to street gang membership. The Revised-G.R.E.A.T program intended to inoculate young people against street gang membership, through the development of skills (i.e., problem-solving, social and communication skills, self-management, and personal responsibility) and creation of achievable goals (Esbensen 2015). A Randomized Control Trial (RCT) evaluation of the Revised-G.R.E.A.T program found, compared to controls, program recipients were 39% less likely to have become a street gang member at one-year follow up (Esbensen et al. 2012), and 24% less likely at four-years follow up (Esbensen et al. 2013). In addition, program recipients demonstrated less anger and expressed more positive attitudes towards law enforcement (Esbensen et al. 2011).

Recently, Growing Against Gangs and Violence (GAGV) has been implemented as a primary prevention measure in the UK, and is provided in areas prioritized in the Ending Gang and Youth Violence initiative (HM Government 2011). Based on G.R.E.A.T, GAGV aims to build young people's resilience towards street gangs and is implemented universally to school year groups. Consistent with the Revised-G.R.E.A.T program, GAGV promotes skill development, whilst also targeting the 'push' (e.g., fear of victimization and peer pressure) and 'pull' (e.g., protection, friendship, and money)

factors associated with street gang membership (see Densley 2018). However, its focus on raising awareness of street gangs and the associated behaviors is closer to the original version of G.R.E.A.T (Esbensen and Osgood 1999).

Outcomes from an RCT found recipients of the GAGV program had 2.72% lower odds of joining a street gang than non-recipients, at a one-year follow-up. However, this did not reach the criteria to be considered statistically significant, meaning findings should be interpreted with caution (Densley et al. 2016). Critically, this may be due to poor retention and attrition rates at the one-year follow-up. Alternatively, as Wong et al. (2011) suggest, primary prevention strategies, such as the original G.R.E.A.T and GAGV, may not be effective at reducing street gang involvement as they are too generic, often failing to target risk factors most strongly related to street gang membership. Despite this, the focus on wellbeing and personal growth, rather than individual blame (Gebo 2016), means primary prevention programs are perceived more positively by communities, schools, and policy-makers than targeted prevention and intervention strategies (Tita and Papachristos 2010). As such, future research needs to consider which risk and protective factors, specific to street gang members, should be targeted in primary prevention strategies.

## 4.2. Secondary Prevention

Although primary prevention strategies should stop the majority of young people from joining street gangs, for those that are not 'immunized' (as coined by the National Gang Center 2020) secondary prevention measures represent the next level in anti-gang strategy. Esbensen (2000) suggests secondary prevention efforts are needed which target young people who have displayed problematic behavior and, as such, are at high risk of joining street gangs. As at-risk youths are most likely to face the decision of whether to join a street gang, secondary prevention programs are often considered the most important strategy in reducing street gang involvement (Howell 2010). Yet, systematic reviews and meta-analyses have failed to find a strong evidence-base supporting the effectiveness of secondary prevention strategies at reducing street gang involvement (Lipsey 2009; Wong et al. 2011).

As highlighted in the "Surveillance" section above, one of the key issues faced in secondary prevention strategies is the accurate identification of young people at risk of street gang involvement. Numerous attempts have been made at creating objective measures to identify youths at high risk of joining a street gang (e.g., Hennigan et al. 2014). However, such instruments often suffer from a lack of predictive validity (Gebo and Tobin 2012). As such, secondary prevention strategies are typically targeted at young people who have had contact with law enforcement due to delinquent behavior or those known to have family members or peers in street gangs (Gebo 2016). Such programs tend to be delivered in areas with high rates of street gangs, as exposure to street gangs is a strong risk factor for membership (Public Safety Canada 2007).

Wyrick (2006) suggests three key elements that any successful secondary prevention program requires. Firstly, at-risk youths need access to alternatives to street gang membership, which are appealing, engaging, and socially rewarding. For potential members, street gangs can be perceived as a source of friendship, excitement, and income (e.g., Augustyn et al. 2019). By diverting at-risk youths' attention onto prosocial alternatives, this will reduce their likelihood of engaging in a street gang. Second, programs need to aid at-risk youths with developing effective support systems. Street gangs offer a source of emotional and social support (Alleyne and Wood 2010). If this support is provided through prosocial relationships, the need to become involved in a street gang will reduce. Finally, Wyrick (2006) stresses that at-risk youths should be held accountable, with clear expectations for appropriate behavior set. As street gang members tend to lack of parental monitoring and discipline (Pedersen 2014), establishing appropriate behaviors in at-risk youths will reduce engagement in street gangs. Due to the sheer number of secondary prevention programs available internationally, examples included in this section are limited to those which have shown some success at preventing street gang involvement, including Cure Violence, Montreal Prevention Treatment Program, Los Angeles Gang

Reduction and Youth Development program, and Functional Family Therapy—Gangs (for an extensive review of street gang prevention programs, see O'Connor and Waddell 2015; Wong et al. 2011).

Los Angeles Gang Reduction and Youth Development (GRYD) is a secondary prevention program designed for young people aged 10–15 years, who are at high-risk of joining a street gang. To be eligible for the GRYD program, young people must exhibit two or more of the following risk factors: antisocial tendencies, weak parental supervision, critical life events, impulsive risk taking, guilt neutralization, negative peer influence, peer delinquency, self-reported delinquency, and familial involvement in a street gang (Brantingham et al. 2017). Using a strengths-based approach, the GRYD program aims to increase resilience towards street gang membership by enhancing protective factors (e.g., support from prosocial peers and family). Evaluation of the GRYD program has had positive results, with reduced engagement in violent and street gang-related behavior at six-month follow-up (Cahill et al. 2015), although this effect was stronger for younger and lower-risk participants, who may be less likely to join a street gang anyway. Critically, evaluations conducted on GRYD failed to include a comparison group of at-risk youths who did not participate in the program, meaning changes in behavior may not be caused by GRYD.

A further secondary prevention program, Cure Violence (formerly CeaseFire), is based on the view that violence is a contagious disease which can be prevented by targeting those most at-risk of 'contracting violence' (Skogan et al. 2009). By identifying and treating high-risk youths, intervening in conflicts and changing community norms, it is assumed that this will reduce engagement in street gangs and the associated violent behavior (McVey et al. 2014). Outcome evaluations of Cure Violence have been mixed; a sixteen-year time series analysis found, after implementation of the program, shootings reduced in five of the seven neighborhoods assessed (Slutkin et al. 2015). However, in one Baltimore neighborhood, violence-related homicides increased by 2.7 times following the implementation of Cure Violence (Webster et al. 2012). The inconsistency in findings may be due to problems with program implementation across different neighborhoods (i.e., poor retainment of staff, lack of consistent funding, communication breakdowns, and limited data sharing; Fox et al. 2015). Having been designed in the USA, where rates of gun violence among street gangs are high, Cure Violence places an inordinate focus on reducing gun-related offending (Butts et al. 2015). As such, Cure Violence lacks generalizability to areas such as the UK, where gun-related violence is low (HM Government 2019).

Recently, researchers have explored whether Functional Family Therapy (FFT), an effective and well-evidenced secondary prevention program typically used for adolescent behavioral and substance misuse problems (Hartnett et al. 2016), could be adapted for young people at-risk of joining a street gang (termed FFT-G). FFT involves treating the family as a whole; working towards establishing better communication, family relationships, and minimizing conflict (Welsh et al. 2014). In FFT-G, issues salient to street gang membership are also targeted (e.g., risk factors, retaliatory behavior, and street gang myths). Outcome evaluations have found young people randomly assigned to receive FFT-G had lower rates of recidivism at 18 months follow-up than the control group (Gottfredson et al. 2018), although, this depended on risk level, with program-recipients at highest-risk of street gang involvement having lower recidivism rates than control, whilst lower-risk program-recipients showed no difference in recidivism rates to the control group (Thornberry et al. 2018). This demonstrates that young people who present with the most risk factors are more likely to benefit from FFT-G. Critically, no research has yet been conducted to examine whether FFT-G is any more successful at reducing street gang involvement than the original FFT program.

The Montreal Preventive Treatment Program (Tremblay et al. 1995) has the longest follow-up period (19 years, with regular follow-ups throughout) of a secondary prevention program (Vitaro et al. 2013). The Montreal Preventive Treatment Program is targeted at boys aged 7–9 years who have displayed disruptive behavior. The program comprises a parental training component (e.g., effective behavioral monitoring, crisis management, and positive reinforcement) and a social skills training component for the child (e.g., self-control skills and building prosocial networks; Tremblay et al. 1991). Evidence from RCTs found that program recipients were less likely to have joined a street gang at both 12 and

15 years-of-age than the control group (McCord et al. 1994; Tremblay et al. 1996). Furthermore, at 24 years-of-age, program recipients were more likely to have graduated from high school and less likely to have a criminal record than the control group (Boisjoli et al. 2007). This demonstrates that secondary prevention programs provided when disruptive behavior first emerges can reduce engagement in street gang membership.

*4.3. Tertiary Prevention*

In situations where primary and secondary prevention programs have not effectively prevented an individual from joining a street gang, tertiary prevention programs can be provided. Tertiary prevention programs target individuals who have already become a street gang member and are aimed at helping them to leave the street gang or making participation in a street gang more challenging (Mora 2020). Typically, tertiary prevention programs are provided to those who are incarcerated or on probation, and have committed an offence related to their street gang membership. However, the provision of tertiary prevention programs is inconsistent, with demand for services far outweighing available resources (Lafontaine et al. 2005; Ruddell et al. 2006). For instance, in the United States alone, it was estimated that 230,000 street gang members were incarcerated in 2011 (National Gang Intelligence Center 2011), meaning the vast majority would not have been able to receive any form of street gang intervention.

Despite this, attempts have been made internationally to develop and implement various tertiary prevention programs for incarcerated street gang members. Typically, prison-based tertiary prevention programs use suppression techniques, such as in-house or legal sanctions for street gang-related behavior and separation from other street gang members. Suppression techniques used to tackle street gang membership are beyond the scope of this paper; for a national analysis see Ruddell et al. (2006). Whilst programs with a therapeutic basis (i.e., providing rehabilitation and support) are offered to a lesser extent in prisons, these are an essential component of a public health approach to street gang membership.

Di Placido et al. (2006) designed a tertiary prevention program for adult street gang members incarcerated in a maximum-security, forensic mental health hospital, which utilized the Risk Need Responsivity (RNR; Andrews et al. 1990) approach to offender rehabilitation. The RNR approach has three key components: (1) risk (treatment intensity should match offenders' risk of recidivism), (2) need (treatment should target criminogenic needs, i.e., factors associated with offending behavior), and; (3) responsivity (treatment style should utilize cognitive social learning methods that are appropriate for each individual offender, accounting for their personal attributes and abilities). In addition, Bonta and Andrews (2007) emphasize professional discretion, whereby clinical judgement can be used to deviate from the previous principles, in exceptional circumstances. The RNR approach is considered the "gold-standard" in offender rehabilitation (Fortune and Ward 2014), with RNR-consistent interventions demonstrating considerable success at reducing recidivism (Andrews and Bonta 2010; Hanson et al. 2009).

At a 24-month follow-up, treated street gang members were less likely to have reoffended violently by 20% and non-violently by 11% than untreated matched controls. In addition, treated street gang members committed fewer major institutional offences than controls. Whilst this program shows promise, the extent to which street gang membership continued post-treatment was not examined; meaning it is not possible to determine whether Di Placido et al.'s (2006) RNR approach is effective at reducing street gang involvement. Furthermore, the RNR approach has been repeatedly criticized for its demotivating nature and limited focus on non-criminogenic needs and therapeutic alliance (Case and Haines 2015; Ward et al. 2007), which are critical factors for providing an effective street gang intervention (Chu et al. 2011; Roman et al. 2017).

A new tertiary prevention program provided in the UK is Identity Matters (IM). Unlike Di Placido et al.'s (2006) program, IM was designed for use in both prison and community settings. IM is targeted at adults whose offending behavior is motivated by identification with a group or street

gang (Randhawa-Horne et al. 2019). Based on Tajfel and Turner's (1986) Social Identity Theory, IM assumes that offending behavior occurs as a result of "over-identification" with the group. Specifically, individuals develop a collective sense of identity based on their group membership. The ingroup is viewed more favorably than outgroups, with group members holding an "us" versus "them" perspective. When social identity is salient, an individual's behavior is guided by group norms (Hogg and Giles 2012). For street gang members, group norms typically include aggressive and violent behavior (Hennigan and Spanovic 2011).

IM consists of 19 structured and manualized sessions which aim to address participants' offence-supportive cognitions, whilst strengthening their sense of personal identity. To date, only one study has been conducted on IM, which consisted of a small-scale process study examining short-term outcomes of a four-site pilot (Randhawa-Horne et al. 2019). Interviews with 20 program completers (14 incarcerated offenders and 6 on probation) were generally positive regarding the content of IM, with the majority recommending no changes. In particular, sessions which explored 'push' (i.e., community disorganization, poverty, unemployment) and 'pull' (i.e., financial gain, status, and protection) factors, desistance, identity, and commitment to change were perceived as most beneficial to participants.

IM was piloted in both a group and one-to-one format. One-to-one sessions were found to be most successful, as participants were more engaged and the program could be tailored to the individuals' needs. However, as discussed previously, demand for IM is high and far outweighs the staffing and time needed to provide the program. Despite this, the safety concerns regarding bringing together members of opposing street gangs for a group-based intervention may overshadow the benefits of increasing recipient numbers. Prison was perceived as the most suitable environment for delivery of IM, with a lack of stability in the community, particularly surrounding accommodation and employment, leading to difficulty in intervention delivery. Pre-post measures showed an increase in participants' understanding of the positive consequences of staying crime-free and negative outcomes from engaging in crime. However, with a lack of control group and small sample size, it is not possible to determine whether the observed changes occurred as a result of engaging in IM. Furthermore, long-term outcome studies need to be conducted to examine whether any changes are maintained post-intervention. Alike Di Placido et al.'s (2006) research, evaluations have not yet been conducted on street gang engagement following receipt of IM; meaning it is not possible to deem this an effective tertiary prevention program.

A number of limitations were highlighted concerning the implementation of IM. Firstly, both facilitators and participants expressed difficulty surrounding the language used in IM. For instance, using the terminology "group", whilst avoiding the term "gang", led to a lack of clarity surrounding the purpose of the intervention. Second, participant motivation was identified as key to intervention success. As street gang members have notoriously poor motivation to engage (Di Placido et al. 2006), interventions should be personally meaningful, positively-oriented, and intrinsically motivating (Fortune 2018). Therefore, the negative orientation of IM (i.e., focusing on harmful past behaviors) is unlikely to improve participants' motivation to engage in the intervention. Third, therapeutic alliance deteriorated throughout the intervention, which is concerning considering past research has consistently demonstrated that a good client-therapist relationship improves the effectiveness of interventions (Gannon and Ward 2014). Fourth, IM is only accredited for use with adult offenders (Ministry of Justice 2020). This is despite the majority of members joining street gangs during adolescence (Pyrooz 2014), which is a period characterized by an increased focus on peer relationships (Young et al. 2014), and high salience of social identity (Tanti et al. 2011). Therefore, an intervention which targets social identity, such as IM, may be more appropriate for young offenders.

Whilst the majority of tertiary prevention strategies are provided in prison settings, as demonstrated in IM these can also be provided in the community. Multi-Systemic Therapy (MST; Henggeler et al. 1992) is a home-based intervention for adolescents, aged 12–17 years, that have engaged in offending behavior (Mertens et al. 2017). According to MST, deviant behavior is a product of the proximal

systems (i.e., family, peer groups, school, and community) that the young person belongs to. As such, MST focuses on risk factors within (e.g., parent-adolescent communication) and between (e.g., parent communication with school) these systems (Henggeler and Schaeffer 2016). As completion of an MST program has been associated with long-term reductions in recidivism (Sawyer and Borduin 2011) and increased contact with prosocial peers (Asscher et al. 2014), it has been recommended as a tertiary prevention program for street gang members (Madden 2013; O'Connor and Waddell 2015).

Findings regarding the effectiveness of MST for street gang members have been mixed. For instance, Boxer et al. (2015) found treatment completion rates were lower for justice-involved youths who self-identified as street gang members (38%), compared to their non-gang counterparts (78%). In particular, street gang members were less engaged in the MST program and were more likely to be removed from the program due to a new arrest (Boxer 2011). Success of MST is partially mediated by reduced contact with delinquent peers (Huey et al. 2000). As ties to a street gang tend to be strong and challenging to break (Decker et al. 2014), it is possible that MST therapists had difficulty reducing the young person's engagement in the street gang (Boxer et al. 2015); reducing overall program effectiveness. Furthermore, street gangs provide access to social and emotional support (Alleyne and Wood 2010), meaning members interpret the street gang as a positive peer network. As MST encourages the formation of positive peer networks, street gang members may be reluctant to leave their street gang (Boxer et al. 2015).

Despite limited support regarding the short-term effectiveness of MST for street gang members, findings examining the longer-term effects have been more positive. Specifically, at one-year follow-up, no difference was found between street gang members and non-gang youths on number of, or time to, re-arrest (Boxer et al. 2017). This suggests that MST appears to have a 'sleeper effect', whereby it is equally effective at reducing recidivism, over a longer time period, in street gang members as non-gang youths. This may be because reducing engagement with a street gang takes time, so changes in behavior will not be seen immediately. However, MST is a relatively novel tertiary prevention program for street gang members, meaning further research is necessary to establish program effectiveness. In general, this section has demonstrated that the evidence-base for tertiary prevention programs is minimal. As such, there is currently no 'gold-standard' approach to intervening with street gang members (Boxer and Goldstein 2012).

## 5. Good Lives Model as a Public Health Framework

The programs reviewed above represent just a small fraction of the wide range of street gang interventions available. Whilst some interventions are emerging as being effective at preventing or reducing street gang involvement, the vast majority suffer from a weak or limited evidence-base. Critically, there is a lack of consistency in the provision of intervention programs for street gang members across communities. Also, Wood (2019) suggests current prevention and intervention strategies are limited by a number of therapeutic issues. Specifically, the benefits of belonging to a street gang (e.g., protection, social and emotional support, sense of identity; Alleyne and Wood 2010) extend beyond the typical proceeds of crime (i.e., financial and material gain), and are not adequately targeted in interventions. In addition, street gang members' mistrust and lack of motivation frequently hinder intervention efforts (Di Placido et al. 2006). The Good Lives Model (GLM; Ward and Brown 2004), a novel approach to offender rehabilitation, can provide a framework for street gang interventions which overcomes these obstacles.

The GLM assumes offending behavior occurs when obstacles prevent the attainment of a meaningful and fulfilling life through prosocial means (Yates et al. 2010). In order to achieve a meaningful and fulfilling life, all humans are naturally predisposed to seek goals fundamental for survival, social networking and reproducing (Laws and Ward 2011). Purvis (2010) proposed 11 universal goals (termed primary goods) which contribute to an individual's wellbeing, happiness, and sense of fulfilment (Ward and Fortune 2013). For a summary of primary goods, see Table 2. Any means necessary and available can be utilized in an effort to attain these primary goods, including both

prosocial and antisocial behaviors. For example, the primary good of Community can be fulfilled through prosocial (e.g., volunteering in the local area) or antisocial methods (e.g., joining a street gang). When antisocial methods are used, it is unlikely that an individual will have a truly meaningful and fulfilling life, as the primary goods are under continuous threat. For instance, street gangs provide members with a sense of safety, protection, and support (Hogg 2014), which are needed to fulfil the primary good of Inner Peace. Yet, at best, Inner Peace will be fulfilled briefly, as street gang membership increases an individual's risk of violent victimization and mental illness (Taylor et al. 2008; Watkins and Melde 2016).

**Table 2.** Eleven Primary Goods and Definitions (Yates et al. 2010).

| | Primary Good | Definition |
|---|---|---|
| 1 | Life | Incorporates basic needs for survival, healthy living, and physical functioning. |
| 2 | Knowledge | Aspiration to learn about and understand a topic of interest (including, but not exclusively, oneself, others, or the wider environment). |
| 3 | Excellence in Work | Pursuing personally meaningful work that increases knowledge and skill development (i.e., mastery experience). |
| 4 | Excellence in Play | Desire to pursue a leisure activity that gives a sense of achievement, enjoyment, or skill development. |
| 5 | Excellence in Agency | Autonomy and independence to create own goals. |
| 6 | Community | A sense of belonging to a wider social group, who have shared interests and values. |
| 7 | Relatedness | Developing warm and affectionate connections with others (including intimate, romantic, and family relationships and friendships). |
| 8 | Inner Peace | Feeling free of emotional distress, managing negative emotions effectively and feeling comfortable with oneself. |
| 9 | Pleasure | Feeling happy and content in one's current life. |
| 10 | Creativity | Using alternative, novel means to express oneself. |
| 11 | Spirituality | Having a sense of meaning and purpose in life. |

Four obstacles have been identified which cause difficulty in obtaining primary goods (Ward and Fortune 2013). Firstly, as discussed above, the use of inappropriate or antisocial means leaves an individual feeling frustrated at their inability to fully secure the primary goods. Second, the primary goods being sought can conflict, or lack coherence, with one another. For example, the primary goods of Community and Excellence in Agency conflict when street gang members focus on group norms, which contradict their personal goals. Third, a lack of scope occurs when primary goods are neglected. For instance, street gang members neglect the primary good of Life (i.e., poor sleep hygiene, lack of routine, reliance on takeaways), in order to spend time with the gang. Fourth, external (e.g., poverty, lack of job opportunities, disorganized neighborhood) and internal obstacles (e.g., impulsivity, low empathy, endorsement of moral disengagement strategies) result in prosocial methods of attaining primary goods being inaccessible.

Critically, the GLM does not specify how to treat street gang members. Rather, it provides a framework which guides the development and implementation of evidence-based interventions (Ward et al. 2011). Specifically, any GLM-consistent intervention should begin by creating a Good Lives Plan, which identifies an individual's skills, the primary goods being sought, and any obstacles they face (for an overview of GLM case formulation, see Fortune 2018). Aligned with a public health approach, GLM-consistent interventions are framed in a manner that promotes well-being, by focusing on achieving personally meaningful goals using prosocial methods (Ward and Fortune 2013). To support the use of prosocial methods, GLM-consistent interventions aim to develop an individual's internal (i.e., skills and values) and external capacities (i.e., resources, support, and opportunities; Ward and Maruna 2007). The GLM framework can guide primary, secondary, and tertiary programs (see Table 3).

**Table 3.** Utilizing a Good Lives Model (GLM) framework for Primary, Secondary, and Tertiary Prevention Programs.

| Stage of Intervention | Overview | GLM Framework |
| --- | --- | --- |
| Primary prevention | Universal prevention programs, provided prior to the onset of street gang membership. | Consistent with the GLM framework, primary prevention programs assist young people (regardless of their risk for street gang involvement) to achieve their primary goods through prosocial means. This involves developing the internal capacity skills necessary for primary good attainment. For instance, school-based programs supporting the development of social skills, goal-making, and emotional competencies can aid in the fulfilment of Relatedness, Excellence in Agency, and Inner Peace. In addition, external obstacles that prevent attainment of primary goods need targeting. For example, mobilizing communities, providing opportunities (e.g., youth groups and employment), and reducing poverty will enable the fulfilment of primary goods through prosocial means. |
| Secondary prevention | Selected prevention programs, targeting individuals who have been identified as at greater risk of joining a street gang. | Utilizing a one-to-one format, secondary prevention programs should begin with a GLM-consistent case formulation. This involves identifying which primary goods are most important to the individual, the means they have available to them, their personal strengths and skills, and any obstacles faced in the pursuit of primary goods (Fortune 2018). This can guide the decision-making process regarding which interventions are most suitable for the individual. For instance, FFT-G will be most appropriate for an individual who is having difficulty attaining the primary good of Relatedness, due to family conflict. Comparatively, an individual who is unable to achieve Inner Peace, because of mental health issues, may respond better to a cognitive-behavioral intervention. As individuals at risk of street gang membership are likely to face obstacles across many of the risk domains (i.e., individual, family, peer, school, and community), a multidisciplinary approach will be necessary to ensure all internal and external obstacles are targeted. |
| Tertiary intervention | Indicated interventions, targeting individuals who have already joined a street gang. | For a street gang member, the perceived benefits of belonging to a street gang (e.g., financial gain, protection, camaraderie), may outweigh the costs (e.g., risk of violent victimization and incarceration). As such, it is important to identify, in case formulation, which primary goods an individual is trying to attain through street gang membership. Again, this informs the selection of appropriate interventions. Tertiary interventions should focus on providing alternative means of achieving the primary goods, without needing to rely on street gang involvement. Similar to secondary prevention programs, this will necessitate a multidisciplinary approach focusing on internal skill development and provision of external resources. Critically, GLM-consistent tertiary interventions must be positively framed; focusing on the strengths and goals of the individual, rather than their risk of returning to the street gang. |

By utilizing a GLM framework, this can enhance existing evidence-based interventions for street gang members. GLM-consistent interventions are strengths-based and goal-focused, which enhances motivation and engagement with the program (Fortune 2018). In addition, as GLM-consistent interventions are positively framed, therapists are encouraged to be empathic and respectful towards clients (Barnao et al. 2015). This supports the development of a strong, trusting therapeutic alliance (Ward and Brown 2004); overcoming issues of high drop-out rates, low therapeutic alliance, and poor client engagement typically seen in street gang interventions. As the GLM has quickly become a favored and widely applied framework for offender rehabilitation internationally (McGrath et al. 2010), using a GLM framework could enable consistency in street gang interventions across communities.

However, as a relatively new framework, empirical evidence regarding GLM-consistent interventions remains in its infancy (Mallion and Wood 2020a; Netto et al. 2014), and is primarily focused on interventions for individuals who have sexually offended (Lindsay et al. 2007; Gannon et al. 2011). Whilst the assumptions of the GLM have been theoretically applied to street gang members (Mallion and Wood 2020b), to date, interventions that are GLM-consistent have not yet been implemented with street gang members. Despite this, the GLM has been successfully applied to young (e.g., Chu et al. 2015; Print 2013; Van Damme et al. 2016) and violent offenders (Whitehead et al. 2007). As street gang members are typically young and engage in violent behavior (Pyrooz 2014; Wood and Alleyne 2010), this supports the use of GLM-consistent interventions with this population.

## 6. Conclusions and Future Directions

There has been a recent shift from viewing street gangs as a problem for law enforcement to considering street gangs as a priority for public health (Catch22 2013). The public health approach emphasizes the role of research in understanding the *causes* of street gang membership, with this informing the development of primary prevention, secondary prevention, and tertiary intervention programs (McDaniel et al. 2014). Whilst research regarding the risk factors for street gang membership has rapidly grown over the past decade, the protective factors preventing involvement are still relatively unknown (McDaniel 2012). As a large number of young people successfully avoid joining street gangs, future research should focus on understanding protective factors which could guide street gang prevention and intervention programs.

A key component of a public health approach involves conducting methodologically sound evaluations of street gang prevention and intervention programs. Whilst this review has demonstrated that some programs are beginning to show promise at reducing street gang involvement (e.g., G.R.E.A.T, FFT-G), the majority of programs lack methodologically sound evaluation (i.e., no control group, reliance on pre-post measures). Furthermore, the use of different definitions of street gang membership across communities has impeded the consistent implementation of prevention and intervention strategies, resulting in mixed findings regarding program effectiveness (e.g., Cure Violence). Thus, to support consistency in the implementation of prevention and intervention programs, it is recommended that the Eurogang definition is used to guide a public health approach to street gangs. Furthermore, in the future, regular evaluations should be embedded into prevention and intervention programs to examine their effectiveness at reducing street gang involvement.

Critically, prevention and intervention programs often suffer from a lack of theoretical foundation and clear goals or objectives (Klein and Maxson 2006; McGloin and Decker 2010). This can be overcome by using the GLM framework to guide evidence-based prevention and intervention strategies for street gang members. The GLM assumes that improving an individual's internal skills and external opportunities will support them in attaining their primary goods through prosocial means. If these primary goods are effectively secured, this will reduce the need for young people to engage with a street gang. As the GLM is a model of healthy human functioning (Purvis et al. 2013), it can be utilized across all stages of prevention and intervention. Whilst past research has theoretically applied the

GLM to street gang members (Mallion and Wood 2020b), future research is needed to empirically examine the application of a GLM framework to street gang prevention and intervention programs.

**Author Contributions:** Conceptualization, J.M. and J.W.; investigation, J.M.; resources, J.M.; data curation, J.M.; writing—original draft preparation, J.M.; writing—review and editing, J.M.; visualization, J.M.; supervision, J.W.; project administration, J.M.; funding acquisition, J.M. and J.W. All authors have read and agreed to the published version of the manuscript.

**Funding:** This research was funded by the Economic and Social Research Council, grant number ES/JS00148/1.

**Conflicts of Interest:** The authors declare no conflict of interest.

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
