# Peer review of "Street Gang Intervention: Review and Good Lives Extension"

_socsci, doi:10.3390/socsci9090160_

Round 1

Reviewer 1 Report

The paper reviews different approaches that have been developed and implemented for addressing gang issues, and then proposes the Good LIves Model for street gang intervention. As there are very few approaches to addressing gang issues that have been found to be highly effective.

The Good Lives Model is an unique and innovative approach that has not been adapted and customized for gang intervention specifically.

The paper is well written, clear, and concise.

The paper wraps up nicely proposing the GLM model after a thorough review of the strengths/limitations of other approaches.

Author Response

Thank you for reviewing our manuscript and the positive comments you gave. As no recommendations were made for revision, we have not changed the manuscript in response to your review. Again, thank you.

Reviewer 2 Report

I want to thank you for the opportunity to review this manuscript. The time spent creating and sending it is greatly appreciated. In addition, I must indicate that I have enjoyed reading it a lot. In my humble opinion, it offers a complete synthesis of the construct, the risk factors involved and the most relevant prevention and intervention programs in street gangs. This type of manuscript, where the various data published over the years in different countries is gathered in a narrative way, to put the current state of the subject on the table is very necessary. Likewise, I want to highlight the narrative structure of the manuscript, which makes it easy to read while expressing itself precisely, which is why I think it would be a very useful document from a scientific and academic point of view. However, I would like to recommend the extension of section 3.2. "Protective Factors for Street Gang Membership". In this section no reference is made to any variable, so it would be appropriate to carry out a review similar to that of the previous point (3.1.) Where the protection factors in the membership of street gangs are collected. Furthermore, it would be interesting if Table 1 were expanded with these new data, including a third column of "protection factors".

Thank you very much for your contribution.

Author Response

Reviewer 2 Revision Recommendations:

I would like to recommend the extension of section 3.2. "Protective Factors for Street Gang Membership". In this section no reference is made to any variable, so it would be appropriate to carry out a review similar to that of the previous point (3.1.) Where the protection factors in the membership of street gangs are collected. Furthermore, it would be interesting if Table 1 were expanded with these new data, including a third column of "protection factors".

Our response:

Thank you for your comments. We have included protective factors that have been identified to date in text. Please see section 3.2, lines 218-227. These are included below for your reference:

“The protective factors that have been identified so far span the individual, family, peer and school domains (for a full summary, see Table 1). Regarding the individual, protective factors for an at-risk young person include having effective coping strategies, high emotional competence and good social skills (Katz & Fox, 2010; Lenzi et al., 2018; McDaniel, 2012). For the family domain, protective factors include strong parental monitoring, cohesiveness within the family and positive parental attachment (Li et al., 2002; Maxson et al., 1998). Interaction with prosocial peer groups is a protective factor within the peer domain (Katz & Fox, 2010). Positive child-teacher relationships, clear familial expectations regarding schooling, and an individual’s commitment to education are all protective factors in the school domain (Stoiber & Good, 1998; Thornberry, 2001). Little is known regarding the protective factors for street gang membership in the community domain.”

We found your suggestion to add a column to our table with protective factors very helpful. We have added these to Table 1 (beginning on line 201). We have directed readers to the table on line 218, with the following:

“(for a full summary, see Table 1)”

Reviewer 3 Report

Well done study. I thoroughly enjoyed reading this manuscript. Well done!

Author Response

(The authors gave the same response as above.)
